# Decoding Hidden Messengers: Proteomic Profiling of Exosomes in Mammary Cancer Research

**DOI:** 10.3390/biomedicines11102839

**Published:** 2023-10-19

**Authors:** Adriana Alonso Novais, Guilherme Henrique Tamarindo, Luiz Gustavo de Almeida Chuffa, Debora Aparecida Pires de Campos Zuccari

**Affiliations:** 1Health Sciences Institute (ICS), Universidade Federal de Mato Grosso (UFMT), Sinop 78550-728, Brazil; aanovais@terra.com.br; 2Brazilian Biosciences National Laboratory, Brazilian Center for Research in Energy and Materials (CNPEM), Campinas 13083-970, Brazil; guilherme.tamarindo@lnbio.cnpem.br; 3Cancer Molecular Research Laboratory (LIMC), Department of Molecular Biology, Faculdade de Medicina de São José do Rio Preto/FAMERP (FAMERP), São José do Rio Preto 15090-000, Brazil; 4Department of Structural and Functional Biology, Institute of Biosciences, Universidade Estadual Paulista (UNESP), Botucatu 18618-689, Brazil; luiz-gustavo.chuffa@unesp.br

**Keywords:** small extracellular vesicles, exosomes, proteomics, signaling molecules, biomarkers, mammary neoplasia

## Abstract

Cancer is a complex and heterogeneous disease, influenced by various factors that affect its progression and response to treatment. Although a histopathological diagnosis is crucial for identifying and classifying cancer, it may not accurately predict the disease’s development and evolution in all cases. To address this limitation, liquid biopsy has emerged as a valuable tool, enabling a more precise and non-invasive analysis of cancer. Liquid biopsy can detect tumor DNA fragments, circulating tumor cells, and exosomes released by cancer cells into the bloodstream. Exosomes attracted significant attention in cancer research because of their specific protein composition, which can provide valuable insights into the disease. The protein profile of exosomes often differs from that of normal cells, reflecting the unique molecular characteristics of cancer. Analyzing these proteins can help identify cancer-associated markers that play important roles in tumor progression, invasion, and metastasis. Ongoing research and clinical validation are essential to advance and effectively utilize protein biomarkers in cancer. Nevertheless, their potential to improve diagnosis and treatment is highly promising. This review discusses several exosome proteins of interest in breast cancer, particularly focusing on studies conducted in mammary tissue and cell lines in humans and experimental animals. Unfortunately, studies conducted in canine species are scarce. This emphasis sheds light on the limited research available in this field. In addition, we present a curated selection of studies that explored exosomal proteins as potential biomarkers, aiming to achieve benefits in breast cancer diagnosis, prognosis, monitoring, and treatment.

## 1. Introduction

In the realm of cancer research, protein biomarkers emerged as powerful tools that unlock a deeper understanding of this complex disease. These molecular signposts found in the body offer invaluable insights into cancer detection, diagnosis, prognosis, and treatment. Their great importance relies on their ability to revolutionize personalized medicine, enhance patient outcomes, and pave the way for targeted therapies. In this review, we delve into the mammary cancer protein biomarkers described for humans and dogs from a proteomic standpoint.

In most of the past four decades and during the most recent years (2010–2019), mammary cancer incidence rates have risen by 0.5% annually [1]. Female breast cancer (BC) exceeds lung cancer statistics (11.7% vs. 11.4%) as the most frequently diagnosed cancer in women, with an estimated 2.3 million new cases yearly, although the mortality rate is much lower (6.9% vs. 18%) [2].

In veterinary medicine, canine mammary tumors (CMT) are the most common cancer in female dogs [3], although they occur mainly in countries where early female sterilization is not a current practice. Due to the high similarity of CMT to human breast cancers, it represents an excellent experimental model for BC. In addition, human BC biomarkers are frequently detectable in cases of CMT [4].

The history of BC is full of attempts to understand the wily nature of this hormone-responsive cancer [5]. A hereditary predisposition may influence screening and follow-up recommendations for high-risk patients, but a specific predisposing gene is identified in less than 30% of cases [6]. In contrast, there is an important role for microenvironmental and epigenetic changes [7].

Tissue biopsy is one of the traditional methods of detecting cancer. However, it is not comprehensive enough to detect the entire genomic scenario of breast tumors. Fortunately, improvements in new techniques, such as liquid biopsy, allowed for the improved management of breast cancer [8].

The liquid biopsy technique uses a sample of tumor cell constituents released from a tumor and/or tumor metastasis in biological substances such as blood, urine, milk, synovial fluid, and saliva. Such constituents may be circulating tumor cells (CTCs), circulating tumor DNA (ctDNA), or circulating tumor RNA (ctRNA), platelets, and exosomes [9]. When we compare liquid biopsy with tissue tumor biopsies, the ease of obtaining bioliquids makes them a very promising source. Furthermore, exosomes released from cancer cells into biofluids carry components that provide important information about tumors [10]. Therefore, liquid biopsies in breast cancers provided encouraging results, especially for monitoring treatment response and predicting disease progression or recurrence [11].

As one of the most prevalent and devastating diseases affecting women worldwide, the early detection and accurate diagnosis of mammary cancer are crucial for improving patient outcomes and implementing effective treatment strategies. Therefore, extensive research efforts are dedicated to identifying reliable biomarkers that can aid in early detection, monitoring, and treatment.

## 2. Breast Cancer Biomarkers

Currently, the histopathological examination of the tumor after surgical excision defines the diagnosis and prognosis of breast cancer. Unfortunately, there are still no reliable histological criteria for metastatic potential before metastasis occurs. There is a great divergence between the histological prognosis and the actual outcome of the disease [12]. The main obstacle to accurate diagnosis is due to the enormous heterogeneity of breast cancer. To overcome this obstacle, several molecular biomarkers have been recently used in clinical practice, such as hormone receptors for breast cancer subtyping and several genes involved in genome conservation, which can predict susceptibility to breast cancer [13,14].

Liquid blood biopsy can detect biomarkers and offers an alternative non-invasive strategy to improve cancer detection [15]. Various biomarkers such as proteins, autoantibodies, miRNAs, nucleic acid methylation, metabolites, lipids, and exosomes demonstrated great potential for detecting cancer, including in the early stages of the disease [15].

Clinical cancer proteomics has been used more recently to identify protein biomarkers and expand the discovery of new potential tumor biomarkers [16]. Furthermore, the evolution of early identification and diagnosis methods is bringing great progress to the treatment of breast cancer, so that each patient and their cancer will be able to receive specific (personalized) therapeutic approaches [17].

## 3. Small Extracellular Vesicles in Cancer Intercellular Communication

Much attention has been given to small extracellular vesicles (SEVs) released by tumor cells for their participation in the transfer of signaling proteins between cancer cells and healthy tissues, which are capable of interfering with their invasive activity [18]. SEVs, also called exosomes, are small membrane nanoparticles released by tissue cells. These small vesicles transport nucleic acids and biologically active proteins (microRNA, mRNA, non-coding RNA, DNA, transcription factors, integrins, signaling molecules, and growth factors) that contribute to cancer progression and metastasis [18,19]. SEVs from tumor cells were demonstrated to preferentially fuse with their resident destination, preparing the niche for metastasis in a process that was intensely investigated by cancer researchers and seems to be highly related to exosomal integrins. Clinical data indicated that exosomal integrins can be used to predict organ-specific metastasis [20,21] (Figure 1).

## 4. SEVs Shuttle Cargo Proteins That Regulate Tumorigenesis and Show Diagnostics and Prognostics Potential

Proteins are among SEV cargo molecules, and they can regulate many processes such as the tropism of cells to receive vesicles, the binding and activation of receptors on recipient cells, and the promotion of some reactions inside the SEVs [22]. However, SEVs do not contain a random array of cargo proteins but rather a specific array. The range of the SEV biocargo reported in the literature was cataloged in EVpedia (http://evpedia.info, accessed on 18 September 2023), Vesiclepedia (http://www.microvesicles.org, accessed on 18 September 2023), and ExoCarta (http://www.exocarta.org, accessed on 18 September 2023) [23].

## 5. Mammary Canine SEV Proteomic Studies Are Scarce

The identification of exosomal proteins involved in tumor signaling through proteomics was recently become a promising field of research. Despite the growing number of studies on exosomal proteomics in humans, particularly in in vitro studies using cell lines, research on the proteomics of exosomes in canine species is still scarce, especially in mammary carcinoma. So far, the literature reported proteins found in tissues and body fluids but not specifically inside small extracellular vesicles.

For example, Klopfleisch and colleagues (2010) [24] used 2D-DIGE and MALDI-TOF-MS and identified 21 proteins with significantly altered protein expression among metastatic canine mammary carcinoma samples. Upregulated proteins included proliferating cell nuclear antigen (PCNA), ferritin light chain (FTL), bomapin, tropomyosin 3 (TPM3), thioredoxin-containing domain containing 5 (TXNDC5), adenosine deaminase (ADA), ornithine aminotransferase (OAT), coronin 1A (CORO1A), (RANBP1) RAN-binding protein 1,3-phosphoglycerate dehydrogenase, and eukaryotic translation elongation factor 1 (eEF1). Downregulated proteins included calretinin, myosin, light chain 2, peroxiredoxin 6, maspin, the ibrinogen beta chain, vinculin, isocitrate dehydrogenase 1, tropomyosin 1, annexin A5, and Rho GTPase activating protein 1. Interestingly, 19 of these 21 proteins were also described in human breast cancer, with their expression associated with malignancy.

One year later, Suárez-Bonnet and colleagues (2011) [25] examined the expression of 14-3-3 σ, a protein related to cell cycle regulation, in normal, dysplastic, and neoplastic canine mammary tissue, to assess the capacity of this protein to act as a marker of myoepithelial cells (MECs). The findings indicated that the 14-3-3 σ protein is present in both normal and neoplastic canine mammary tissue, with high expression of this antigen in MECs. These results suggest that this protein may have a role in enhancing the spreading capacity of canine mammary tumors and can be considered both a prognostic biomarker and a therapeutic target.

Later, Jagarlamudi et al. (2014) [26] analyzed a cell-cycle-related protein called serum thymidine kinase 1 (sTK1), reporting that its levels and activity were significantly higher in CMT than in healthy dogs.

Other authors [27], in 2019, reported the detection of the biomarker BIRC5 (survivin) in dog serum, in primary culture of canine mammary tumor cells and in the canine mammary cancer cell line REM-134. They found that serum levels of this protein, which is involved in inhibiting cell death, were 109.83 ± 8.10 pg/mL in dogs with CMT, compared with 44.7 ± 2.61 pg/mL and 30.23 ± 1.32 pg/mL, respectively, in dogs with non-cancerous diseases and healthy individuals.

Fortunately, school attendance improved afterward. Fhaikrue and colleagues (2020) [28] developed a study to identify protein expression in canine mammary tumors (CMTs) using primary cell cultures from benign mixed tumors, simple carcinomas, complex carcinomas, and healthy mammary glands using a proteomic approach. Cytokeratin 5 (CK5) and transketolase (TKT) were identified in benign mixed tumor cells and complex carcinoma cells. Furthermore, cytokeratin 18 (CK18) and pyruvate kinase (PKM) were identified in simple carcinoma cells. Furthermore, they identified the tumor antigen alpha-2-HS-glycoprotein in complex carcinoma cells specifically, whereas ATP-dependent platelet-like proteins 6-phosphofructokinase and elongation factor 2 proteins were observed in benign cells. These changes in genes related to metabolism indicate that MTCs undergo metabolic reprogramming that transforms benign tumors into malignant ones, meeting the demands of proliferating cells [29].

In the same year, Park and colleagues (2020) [30] studied the proteomics of plasma from healthy dogs and dogs with cancer, focusing on uncovering biomarkers of aggressiveness of canine mammary tumors. They identified 54 proteins that were higher in cancer than in normal plasma, including SERPING1, SERPINA6, and lecithin cholesterol acyltransferase (LCAT). The authors concluded that the plasma protein LCAT could be considered a biomarker for advanced breast cancer and metastasizing breast tumors.

Shortly thereafter, Cordeiro and collaborators (2021) [31] compared the proteomic profile of canine tumor cells at different degrees of differentiation. The authors demonstrated that the malignant phenotype could be a consequence of changes in the expression of key proteins such as FNDC1, A1BG, CANX, HSPA5, and PDIA3, which could lead to tumor evasion against inflammatory cells, thus facilitating the spread of cancer.

Additionally, Yuan and colleagues (2021) [32] studied anterior gradient protein 2 (AGR2), a chaperone and p53 inhibitor involved in cell migration, transformation, and metastasis. They described that it was overexpressed in tissue samples from canine mammary malignant tumor (MMT) tissues and that high levels of AGR2 in sera from dogs with MMT were associated with the progression and remote metastasis of MMT and a low overall survival rate.

As previously documented, research on mammary tumor proteins in the exosomes of dogs is currently non-existent. As a result, there is a pressing need for further investigation in this area. Despite extensive research conducted on mammary tumors in dogs, there was relatively little focus on the proteins in the exosomes derived from these tumors. Therefore, the comprehensive characterization and exploration of protein cargo within dog mammary tumor exosomes remain largely unexplored.

## 6. SEV Proteomics May Identify Novel Breast Cancer Biomarkers

Extensive efforts have been dedicated to identifying reliable biomarkers that can aid in the early detection and monitoring of human mammary cancer. The proteomic analysis of exosomes has gained significant attention in mammary cancer research because of the abundance and diversity of proteins encapsulated within these vesicles. By characterizing the protein composition of exosomes derived from mammary cancer cells, researchers can uncover unique molecular signatures specific to cancerous cells.

In recent decades, advances in proteomic analysis have presented the main objective of discovering molecular biomarkers for the early detection of cancer, the characterization of the tumor profile, and the identification of new therapeutic targets [33]. A particular advantage of the proteome is that not only tissues but also body fluids such as blood, urine, and saliva can be used to investigate the molecular correlation between disease and drug action [34,35].

For example, Khan and colleagues (2014) [36] demonstrated that the protein survivin and its splice variants were present in the cargo of exosomes isolated from the serum of breast cancer patients, mimicking a pattern they had also reported in breast cancer tissue samples. They concluded that differential expression of exosomal survivin, particularly survivin-2B, could serve as a diagnostic and/or prognostic marker in patients with early-stage breast cancer (Table 1).

Harris and colleagues (2015) studied the proteomic profiling of exosomes released from three breast cancer cell lines (MCF-7, MDA-MB-231, and Rab27b) and identified 85 differentially expressed proteins. In metastatic-tumor-derived exosomes, they observed upregulation of a unique set of adhesion proteins (vimentin, galectin-3-binding protein, annexin A1, plectin, protein CYR61, EGF-like repeat and discoidin I-like domain containing protein, filamin-B, and protein-glutamine gamma-glutamyltransferase 2) [37].

Using proteomic analysis, Blomme and colleagues (2016) [38] validated a novel exosomal protein, termed myoferlin, which is related to angiogenesis, metabolism reprogramming, and epithelial–mesenchymal transition in cancer [39]. They demonstrated that myoferlin depletion in cancer cells leads to exosomes that are functionally deficient and have a significantly reduced ability to induce the migration and proliferation of those cells.

Vardaki and colleagues (2016) [40] documented significant differences not only in the number of exosomes secreted but also in the protein content of exosomes secreted by metastatic vs. non-metastatic tumors. The authors identified periostin, a protein that appears to bind to integrins in cancer cells, triggering the Akt/PKB and FAK signaling pathways and increasing angiogenesis, invasion, metastasis, and cell survival [41]. The presence of periostin was validated in a pilot cohort of samples from breast cancer patients with localized disease or lymph node metastasis.

Hurwitz and colleagues (2016) [42] used sixty National Cancer Institute cell lines (NCI-60) to provide the largest proteomic profile of SEVs in a single study, identifying a total of 6071 proteins. Only tetraspanins CD81, Alix, and HSC70 were found in all samples. The periostin protein was confirmed in two metastatic breast cancer cell lines (MDA-MB-231 and HS 578T) but was not detected in other non-metastatic breast cancer SEVs. Other proteins, such as ratilin, fibulin-7, and plasminogen activator inhibitor 1, were exclusively found in SEVs from metastatic breast cancer. The authors concluded that SEVs reliably represent their progenitor cells and are excellent candidates as biomarkers for cancer diagnosis, progression, and metastasis.

Moon et al. (2016) [43] identified fibronectin (FN), a protein that mediates the interaction of cells with the extracellular matrix (ECM) and downstream factor that promotes metastasis [44], as a biomarker candidate because of its presence on the surface of SEVs secreted from human BC cell lines. FN levels were significantly elevated at all stages of BC and returned to normal after tumor removal. At early stages of BC, another study conducted by Lee and colleagues (2017) [45] reported Developmental Endothelial Locus-1 (DEL-1) as a possible diagnostic tool to distinguish benign tumors from a healthy breast.

Gangoda et al. (2017) [46] compared exosomes isolated from several strains of genetically related mouse breast tumors with different metastatic propensities through proteomic analysis. The authors observed that the exosomes derived from metastatic cells were rich in proteins capable of promoting cell migration, proliferation, invasion, and angiogenesis. On the other hand, exosomes derived from non-metastatic cells contained proteins involved in cell–cell/cell matrix adhesion and polarity maintenance. The contents of metastatic exosomes revealed membrane proteins, including ceruloplasmin and metatherin, which could help target primary cancer cells to specific metastatic sites.

The annexin A2 protein was reported by Maji et al. (2017) [47] to be important for signaling in the breast cancer microenvironment, promoting angiogenesis and vascularization. Furthermore, annexin A2 induced the activation of macrophages, favoring breast cancer metastasis in distant organs. The authors concluded that exosomal annexin A2 may be a potential biomarker and therapeutic target for the diagnosis and treatment of metastatic breast cancer.

Rontogianni and colleagues (2019) [48] focused on defining EV-subtype-specific signatures that could play a role in non-invasive diagnostic testing. To this end, they profiled the proteomes of SEVs secreted by BC cell lines and patient serum, with a special emphasis on the TNBC and HER2 subtypes. Some representative TNBC-signature proteins included ephrin type-A receptor 2, DnaJ homolog subfamily A member 1, polyadenylate-binding protein 1, and neuropilin-1, which showed higher expression levels in the patient’s SEVs compared with the SEVs of HER2-positive patients. Similarly, receptor tyrosine protein kinase erbB-2, growth factor receptor-bound protein 7, eukaryotic translation initiation factor 3 subunit H, and brefeldin A-inhibited guanine nucleotide-exchange protein 2 were the most discriminative protein markers for serum-derived SEVs from HER2-positive patient. Their data revealed very distinct proteomic profiles across the different cell-line-derived SEVs, thus reflecting the unique biology of their breast cancer subtype.

The distinct proteomic content of SEVs was also demonstrated in invasive breast cancer cell lines compared with noninvasive breast cancer cells. SEVs produced by invasive MDA-MB231 cells were significantly enriched for proteins involved in vesicle formation, protein synthesis, proteolysis, and glycolysis. Conversely, SEVs produced by MCF10 were significantly enriched in membrane proteins, adhesion molecules, proteins involved in cellular migration, and components of the extracellular matrix (ECM). Based on these differences, the most abundant proteins uniquely identified in MDA-MB231 SEVs were those involved in transcriptional regulation (spliceosome, transcription factors, ribosomal proteins, and tRNA ligases), proteolysis (proteasome units abd pyrophosphatase), EV formation (annexin and vesicle markers LAMP-1 and EEA1), cell cycle (NUMA1), cell motility, and adherence to extracellular matrices (vitronectin, collagen, filamin proteins, and EDIL3) [20].

While examining the protein content of SEVs derived from three breast cancer cell lines (MCF-7, MDA-MB-231, and T47D), Dalla and colleagues (2020) [49] found that seven proteins were the most abundant (actin cytoplasmic 1, pyruvate kinase, glyceraldehyde-3-phosphate dehydrogenase, 60 kDa mitochondrial heat shock protein, mitochondrial ATP synthase alpha subunit, sodium/potassium transport ATPase beta-3 subunit, and voltage-dependent anion selective protein channel 2). They concluded that once the majority are proteins related to the metabolism of mitochondrial processes, their presence supports the hypothesis of the relevant role of SEVs in metabolic reprogramming.

Complementing previous studies, Risha and colleagues (2020) [50] studied SEVs’ proteomes from MDA-MB-231 and MCF-10A cell lines. They reported that 16 proteins were mainly involved in the formation of cancer metastasis. Among them, three exosomal membrane/surface proteins, glucose transporter 1 (GLUT-1), glypican 1 (GPC-1), disintegrin, and metalloproteinase domain-containing protein 10 (ADAM10), were identified as potential breast cancer biomarkers. The authors also concluded that SEVs can mediate distinct molecular mechanisms such as glucose uptake and ECM remodeling.

In the same year, Vinik and colleagues (2020) [51] isolated small fractions enriched in SEVs from the plasma of healthy controls and BC patients at different stages of the disease, before and after surgery. Proteomic analysis revealed a signature of seven proteins that differentiated BC patients (fibronectin, FAK, MEC1, B-actin, p90RSK_pT573, N-Cadherin, and C-Raf). Among them, FAK and fi-bronectin demonstrated high diagnostic accuracy.

The study on the proteomics of serum exosomes derived from 10 TNBC patients and 17 healthy donors by Li and colleagues (2021) [52] found that the expression levels of tetraspanin CD151 in TNBC-derived exosomes were significantly higher than those in healthy patient exosomes, validating their findings in 16 additional donor samples. The authors also observed that exosomal CD151 facilitated the secretion of ribosomal proteins while inhibiting the exosomal secretion of complement proteins. Notably, CD151-deleted exosomes significantly decreased the migration and invasion of TNBC cells.

In a different approach, the scientific team of Patwardhan and colleagues (2021) [53] demonstrated the role of exosomes in the stiffness of the ECM, triggering the invasiveness of breast cancer. Proteomic analysis of exosomal lysates revealed the enrichment of cell adhesion and migration proteins in exosomes from rigid ECM cultures compared with those from soft cultures. Thrombospondin-1 (THBS1) was identified as a prospective regulator of ECM stiffness-dependent cancer invasion involving matrix metalloproteinase and focal adhesion kinase.

Taken together, these data may support the development of new diagnostic tools, but further investigation is still required (Table 1).

**Table 1 biomedicines-11-02839-t001:** List of main SEV-associated proteins as potential biomarkers for BC according to publications from 2014 to 2021. Proteins highlighted in red mean upregulated, while those in blue mean downregulated.

Author	Proteins	Isolation	Pathology	Applicability	Source
Khan et al. (2014) [36]	**BIRC5** and **splice variants**	UC	TNBC, ER, and PR positive	Diagnosis, prognosis, and treatment	Tumor
Harris et al. (2015) [37]	Several proteins related with **adhesion/motility/cytoskeleton**, **proteases**, **transporters**, **cell surface receptor**, **stress response proteins**, **small GTPases**, **metabolic enzymes**, and **RNA binding**	UC	TNBC	Treatment	Cells in vitro
	Several proteins related with **tetraspanin**, **adhesion**, **cell surface receptor**, **transporter**, **stress response proteins**, **budding vesicles**, **trafficking/transport**, **calcium binding**, and **small GTPase**	UC	Luminal		
Blomme et al. (2016) [38]	**MYOF**	UC	Luminal and TNBC	Treatment	Cells in vitro
Vardaky et al. (2016) [40]	**POSTN**	UC	TNBC cells and patients with LN metastasis	Diagnosis	Cells in vitro and tumor samples
Hurwitz et al. (2016) [42]	**POSTN**	ExtraPEG and UC	TNBC	Diagnosis	Cells in vitro
	**RFTN1**, **FBLN7**, and **SERPINE1**		Metastatic breast cancer		
Moon et al. (2016) [43]; Lee et al. (2017) [45]	**DEL-1**	UC	Early-stage breast cancer, several subtypes	Diagnosis	Cells in vitro and plasma samples
Gangoda et al. (2017) [46]	**CP **and **MTDH**	UC	TNBC, highly metastatic	Diagnosis	Cells in vitro
Maji et al. (2017) [47]	**ANXA2**	UC	Metastatic breast cancer cells	Diagnosis and treatment	Cells in vitro
Rontogianni et al. (2019) [48]	**EPHA2,** **DNAJA1**, **PABPC1**, and **NRP1**	UC	TNBC	Diagnosis	Cells in vitro and serum samples
	**HER2**, **GRB7**, **EIF3H**, and **ARFGEF2**		HER2		
Jordan et al. (2020) [20]	Several proteins related with **spliceosome**, **transcription factors**, **ribosomal proteins**, **tRNA ligases**, **proteasome units**; **pyrophosphatase**, **annexin**, **LAMP-1**, **EEA1**, **NUMA1**, **VTN**, **collagen**, **filamin proteins**, and **EDIL3**	Sepharose CL-2B SEC	TNBC	Diagnosis	Cells in vitro and serum samples
	Proteins from **adherin family members**, **laminin proteins**, **proteoglycans**, **SDC1**, **EPCAM**, **b-catenin**, **collagen**, **CD109**, **RARRES1**, **PTGFRN**, **FAT1**, **S100A14**, **AREG**, **calcium-binding proteins**, **serine proteases**, and **cholesterol-** and **lipoprotein-binding proteins**		Ductal carcinoma in situ		
Dalla et al. (2020) [49]	**Actin cytoplasmic 1, PKM, GAPDH**,** HSP60**,** ATP1B3**, and **VDAC2**	UC	Shared among MCF-7, MDA-MB-231, and T47D	Diagnosis	Cells in vitro
Risha et al. (2020) [50]	**GLUT1**, **GPC3** and **ADAM10**	UC	MDA-MB-231	Diagnosis and prognosis	Cells in vitro
Vinik et al. (2020) [51]	**EGFR**, **FAK**, **fibronectin**, **p38_pT180_Y182**, **N-cadherin**, **E2F1**, **PARP**, **MEK1**, **Aurora-B**, **p90RSK_pT573**, **S6_pS240_S244**	SEC	Stage I	Diagnosis, and treatment	Serum samples
	**C-Raf**, **fibronectin**, **heregulin**, **FAK**, **MEK**, **β-actin**, **N-cadherin**, **FoxO3a_PS318_S231**, **P-cadherin**, **PDHK1**, **TAZ**		Stage IIA		
Li et al. (2021) [52]	**CD151**	UC	TNBC	Diagnosis	Serum samples
Patwardhan et al. (2021) [53]	**THBS1**	ExoEnrich	TNBC	Treatment	Cells in vitro

Legend: BIRC5—survivin; MYOF—myoferlin; POSTN—periostin; FBLN7—fibulin 7; RFTN1—raftilin; SERPINE1—plasminogen activator inhibitor 1; DEL-1—Developmental Endothelial Locus-1; CP—ceruloplasmin; MTDH—metadherin; ANXA2—annexin A2; EPHA2—EPH receptor A2; DNAJA1—DnaJ Heat Shock Protein Family (Hsp40) Member A1; PABPC1—Poly(A) Binding Protein Cytoplasmic 1; NRP1—Neuropilin 1; HER2—Human Epidermal Growth Factor Receptor 2; GRB7—Growth Factor Receptor-Bound Protein 7; EIF3H—Eukaryotic Translation Initiation Factor 3 Subunit H; ARFGEF2—ADP Ribosylation Factor Guanine Nucleotide Exchange Factor 2; VTN—Vitronectin; EDIL3—EGF Like Repeats And Discoidin Domains 3; SDC1—Syndecan-1; EPCAM—Epithelial Cell Adhesion Molecule; RARRES1—Retinoic Acid Receptor Responder 1; PTGFRN—Prostaglandin F2 Receptor Inhibitor; FAT1—FAT Atypical Cadherin 1; S100A14—S100 Calcium Binding Protein A14; AREG—Amphiregulin; PKM—pyruvate kinase; GAPDH—Glyceraldehyde-3-Phosphate Dehydrogenase; ATP1B3—ATPase Na+/K+ transporting subunit beta 3; VDAC2—Voltage-Dependent Anion-Selective Channel Protein 2; HSP60—60 kDa heat shock protein, mitochondrial; GLUT1—glucose transporter 1; GPC3—Glypican 1; ADAM10—disintegrin and metalloproteinase domain-containing protein 10; THBS1—Thrombospondin-1; EGFR—Epidermal Growth Factor Receptor; FAK—focal adhesion kinase; E2F1—E2F Transcription Factor 1; PARP—Poly(ADP-Ribose) Polymerase; MEK1—Mitogen-Activated Protein Kinase Kinase 1; PDHK1—Pyruvate Dehydrogenase Kinase 1; TAZ—WW Domain Containing Transcription Regulator 1; CD151—Platelet-Endothelial Tetraspan Antigen 3; UC—Ultracentrifugation; SEC—size-exclusion column.

## 7. Concluding Remarks and Perspectives

In view of the possibility of performing early diagnosis and longitudinal prognostic evaluation, the number of studies focusing on the exosomal proteomics of patients diagnosed with breast tumor has grown. In addition, one of the most promising applications of cancer protein biomarkers relies on the development of targeted therapies. By identifying specific proteins that drive tumor growth, metastasis, or resistance to treatment, researchers can design drugs or treatment strategies that specifically target these biomarkers. In addition, the identification of a group of proteins allows us to clarify what type of biological processes are deregulated due to SEVs. In the present review, it is clear that metabolic reprogramming occurs not only at distinct points but also in several proteins that mediate cell interaction with ECM, suggesting the key role of SEVs in metastasis. This could improve advances in the personalized approach, known as precision medicine, which maximizes treatment effectiveness while minimizing side effects on healthy tissues. For obvious reasons, research in the canine species is still scarce, albeit important because CMT is considered as an excellent experimental model for BC. Due to the shorter lifespan of dogs and the rapid progression of CMTs, researchers can gain insights into tumor development, metastasis, and therapeutic responses in a relatively shorter timeframe compared with human studies. On the other hand, liquid biopsy has several advantages over traditional tissue biopsy in the context of breast cancer because it enables the identification and characterization of tumor-specific genetic alterations and mutations, even at early stages when traditional imaging techniques may not detect the disease. Its non-invasive nature makes it an attractive option for serial sampling, thereby reducing patient discomfort and facilitating longitudinal studies. Therefore, unveiling biomarker proteins carried by exosomes through liquid biopsy may provide valuable information. Indeed, if biomarker detection occurs before tumor signaling, there is a chance to interrupt the pathogenesis of the disease. Biomarker proteins can be modulated at various levels, such as by silencing genes, affecting their transcription or even the protein signaling pathway. It would be possible, for example, to block the signaling effect of exosomal metastasis-inducing proteins or even to modulate this effect by introducing protective signaling proteins into the vesicles. Cancer protein biomarkers revolutionized cancer research and patient care, ushering in a new era of precision medicine. As researchers continue to unravel the intricate landscape of cancer biology, protein biomarkers will remain invaluable in the fight against cancer, bringing hope for improved outcomes, enhanced quality of life, and, ultimately, a world free from the burden of this devastating disease.

## Figures and Tables

**Figure 1 biomedicines-11-02839-f001:**
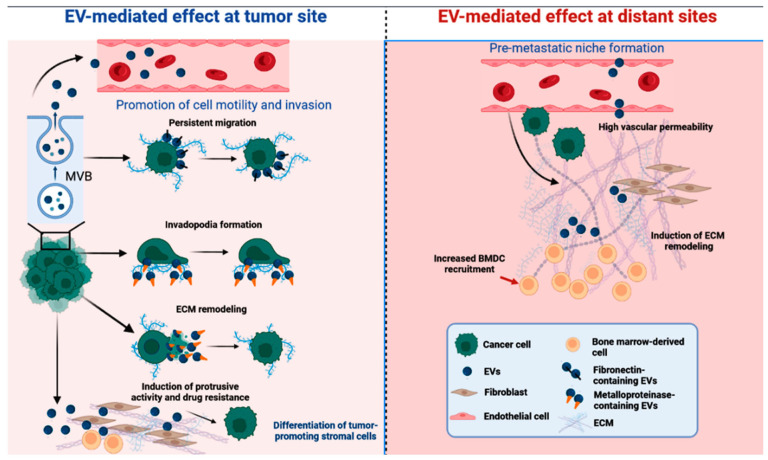
Diverse cellular signaling of exosomes (SEVs) was described to influence the establishment and spread of cancer. EV-mediated effects may occur at tumor sites (e.g., persistent migration, invadopodia formation, ECM remodeling, induction of protrusive activity and drug resistance, and differentiation of tumor-promoting stromal cells) and at distant sites (e.g., high vascular permeability, induction of ECM remodeling, and increased BMDC recruitment) [19]. EV, extracellular vesicle; MVB, multivesicular bodies; ECM, extracellular matrix; BMDC, bone-marrow-derived cell.

## Data Availability

Data sharing is not applicable.

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
