# Peer review of "Decoding Hidden Messengers: Proteomic Profiling of Exosomes in Mammary Cancer Research"

_biomedicines, 2023, doi:10.3390/biomedicines11102839_

Round 1
Reviewer 1 Report
The authors have composed a manuscript named “Decoding the Hidden Messengers: Proteomic Profiling of Exosomes in Mammary Cancer Research”. This manuscript presents a compilation of exosomal proteins that have been previously identified as potential biomarkers for breast cancer diagnosis, prognosis, or treatment in various studies. However, there are several substantial issues that require attention and correction before the manuscript can be considered for publication These issues are outlined below:
1. The manuscript conflates the terms "extracellular vesicles" (EVs) and "exosomes." It is crucial to adhere to the recommended classification provided by the "Minimal Information for Studies of Extracellular Vesicles 2018" (MISEV2018) guidelines, which categorize EVs into small extracellular vesicles and medium/large extracellular vesicles. The authors should make the necessary revisions to ensure that the correct terminology is used throughout the manuscript.
2. Accurate proteomic analysis of EVs necessitates rigorous purification methods to obtain high-quality samples. The authors should include a comprehensive description of the purification techniques employed, such as ultracentrifugation and tangential flow filtration.
3. Also, for breast cancer liquid biopsy, it is important to acknowledge the potential utility of alternative biofluids beyond blood, such as milk and urine. Discussing the advantages and challenges associated with different biofluids would provide a more comprehensive perspective on liquid biopsy approaches.
4. The layout and content of the summary table need to be reorganized to enhance clarity and utility. Specifically, the table should indicate whether the identified biomarkers are upregulated or downregulated in breast cancer. Furthermore, the table should clearly specify whether these biomarkers are associated with diagnosis, prognosis, or potential treatment strategies. Lastly, it should provide information on the cellular sources of the exosomes containing these biomarkers. This restructuring will make the table more informative and accessible to readers.
Author Response
The authors thank the reviewer for all valuable comments and suggestions to improve the quality of our manuscript.
Please, find the point-by-point file attached.

Reviewer 2 Report
In their concise review, the authors have adeptly highlighted recent developments in breast cancer research, placing particular emphasis on the less commonly utilized canine models of metastasis. They have also provided a comprehensive illustration of the effects mediated by extracellular vesicles (EVs) in the vicinity of the tumor as well as at distant sites.
The reviewer appreciates the summary of a tumor model in larger animals, as opposed to the more conventional mouse and rat models.
However, there is one issue that requires correction:
Line 48: "accounting for 1 in 4 cancer cases and for 1 in 6 cancer deaths." Reference 2, as shown in Figure 4A, indicates that the overall incidence of female breast cancer is 11.7%, with a mortality rate of 6.9%. It appears that the authors are referring to Figure 4C, which pertains specifically to females. Therefore, it is necessary to rectify this error.
Author Response

(The authors gave the same response as above.)

Round 2
Reviewer 1 Report
The manuscript is accepted for publishing.